# Evaluation of Dalbavancin Use on Clinical Outcomes, Cost-Savings, and Adherence at a Large Safety Net Hospital

Richard Lueking,[a] Wenjing Wei,[b] Norman S. Mang,[a] Jessica K. Ortwine,[a] Jessica Meisner[c]

[a]Division of Infectious Diseases and Geographic Medicine, Department of Medicine, University of Texas Southwestern, Dallas, Texas, USA
[b]Department of Pharmacy Services, Parkland Health, Dallas, Texas, USA
[c]Division of Infectious Diseases, Department of Medicine, University of Pennsylvania Perelman School of Medicine, Philadelphia, Pennsylvania, USA

**ABSTRACT** Dalbavancin is a second-generation lipoglycopeptide antibiotic with activity against Gram-positive organisms. Dalbavancin is Food and Drug Administration (FDA)-approved for acute bacterial skin and soft tissue infections (ABSSTIs). There is a lack of substantial data on dalbavancin in more invasive infections, particularly in high-risk populations (patients with intravenous drug use and unstable living conditions). In this retrospective observational study, we reviewed all patients that received at least one dose of dalbavancin in an inpatient or outpatient setting at Parkland Hospital from February of 2019 to August of 2021. The demographics, type of infection, and rationale for dalbavancin were collected at the baseline. Clinical failure was measured by an avoidance of emergency department (ED) visits or hospital readmission at 30, 60, and 90 days. A separate analysis was conducted to estimate hospital, rehabilitation, or nursing facility days saved based on the projected length of treatment. 40 patients were included, and the majority were uninsured (85%), experiencing homelessness (60%), or had intravenous drug use (IDU) (57.5%). Indications for use included ABSSTIs (45%), bloodstream infection (67.5%), osteomyelitis (40%), infective endocarditis (10%), and septic arthritis (10%). Clinical failure was observed in 5 of the 40 patients (12.5%). Nonadherence to medical recommendations, a lack of source control, and ongoing IDU increased the risk of failure. Dalbavancin saved a total of 566 days of inpatient, rehabilitation, and nursing facility stays. Dalbavancin is a reasonable alternative to the standard of care in an at-risk population, offering decreased lengths of stays and cost savings. The uses of second-generation lipoglycopeptides are desirable alternatives to traditional outpatient parenteral antibiotic therapies for patients who otherwise would not qualify or for patients who desire less hospital contact in light of the COVID-19 pandemic.

**IMPORTANCE** This study contributes additional experience to the literature of dalbavancin use in off-label indications, particularly for patients who do not qualify for outpatient parenteral antimicrobial therapy. The majority of the patient population were people who inject drugs and the uninsured. There is difficulty in tracking outcomes in this patient population, given their outpatient follow-up rates; however, we were able to track emergency room visits and readmissions throughout the majority of the local metroplex. The clinical use of dalbavancin at our institution also increased in the midst of the COVID-19 pandemic in an effort to preserve hospital resources and limit health care exposure. In addition, we are able to provide institution-specific cost-saving data with the use of dalbavancin.

**KEYWORDS** dalbavancin, antibiotics, Gram-positive bacteria, infectious disease, lipoglycopeptides

Dalbavancin is a long-acting, second-generation, lipoglycopeptide antibiotic with potent activity against Gram-positive organisms. Dalbavancin is currently Food and Drug Administration (FDA)-approved for acute bacterial skin and soft tissue

Address correspondence to Richard Lueking, richard.lueking@gmail.com, or Jessica Meisner, jessica.meisner@pennmedicine.upenn.edu.

The authors declare no conflict of interest.

infections (ABSSTIs) (1–2). Growing evidence suggests that patients can be successfully treated with dalbavancin for indications outside of skin and soft tissue infections, including bloodstream infections (BSI) and osteomyelitis (OM) with significant cost savings and reduced lengths of stays (3–8). The use of dalbavancin is potentially important for patients who are not candidates for outpatient parenteral antibiotic therapy (OPAT) due to their unstable living conditions or intravenous drug use (IDU). The use of dalbavancin is also potentially important in minimizing hospital and nursing home admissions as well as health care exposure, especially in light of the COVID-19 pandemic.

The most desirable quality of dalbavancin is its long half-life, which results in the ability of 1 or 2 doses to constitute an entire therapeutic course. This extended half-life of approximately 14.4 days is primarily due to its high protein-bound nature, and it also has the addition of an extended, lipophilic side chain which more effectively anchors the molecule to the cell wall (9–10). Pharmacokinetic (PK) modeling has predicted that a regimen of 1,500 mg of dalbavancin on day 1 and day 8 will achieve adequate concentrations in bone and articular tissue and will, in addition, show adequate serum concentrations so as to achieve pharmacodynamic (PD) targets against the majority of *Staphylococcus aureus* isolates (11). Few randomized control trials exist for indications outside ABSSTIs and OM (4). There are limited studies on the use of dalbavancin in high-risk patients, including people who inject drugs (PWID) and those with unstable living situations (7, 12–14).

The purpose of the study was to assess the baseline characteristics and clinical outcomes of all patients treated with dalbavancin for any indication at a large safety net hospital over a 25-month period, which included 18 months of the COVID-19 pandemic. The patient population was unique in comparison to other, earlier studies evaluating dalbavancin use in that a majority of patients had unstable living conditions and had concomitant IDU. This study includes patients receiving treatment for ABSSTIs, BSI, OM, infective endocarditis (IE), and septic arthritis due to suspected or confirmed Gram-positive infections.

## RESULTS

**Patient characteristics.** A total of 40 patients were included (26 males and 14 females), with a median age of 49 (interquartile range [IQR] of 16.25). All patients were greater than 18 years of age. The majority of subjects were PWID, uninsured, and had unstable living conditions. Only 47.5% of the patients underwent procedural/surgical source control. Other baseline characteristics, including comorbidities, are shown in Table 1.

**Indications for use and pathogens.** Clinical indications for dalbavancin use included ABSSTIs, BSIOM, IE, and septic arthritis. 23 (57.5%) patients had more than one clinical indication. All included patients had a microbiologically confirmed Gram-positive infection, with the most frequently isolated pathogens being methicillin-resistant *Staphylococcus aureus* (MRSA) ($n = 23$, 57.5%), methicillin-sensitive *S. aureus* (MSSA) ($n = 12$, 30%), *Streptococcus* species ($n = 7$, 17.5%), coagulase-negative *Staphylococcus* (CoNS) ($n = 5$, 12.5%), and Enterococcus species ($n = 1$, 2.5%). 11 (27.5%) patients had polymicrobial Gram-positive infections. The clinical indications and causative pathogens are shown in Table 2.

**Dalbavancin therapy and adverse drug events.** A total of 3 dosing strategies were noted in this review. 27 patients received a 1,500 mg intravenous infusion as a single dose, 1 patient received a renally adjusted dose of 1,125 mg as a single infusion, and 12 patients received 1,500 mg intravenous infusions on day 1 and day 8. A total of 52 total dalbavancin doses were administered, 13 (25%) of which were given in an outpatient setting. 10 patients received at least 1 dose in an outpatient setting, 4 of which were treated entirely as outpatients in the infectious diseases (ID) clinic (Table 2).

Only 2 of the 40 patients had an identified adverse drug event. Events were recorded either during their active admission or during their infusion visits in the ID clinic. One patient developed *Clostridioides difficile* colitis, which was treated with oral vancomycin, during admission, and the other had sudden onset substernal chest pain

**TABLE 1** Demographics and comorbidities

| Characteristics[a] | Value (N = 40) |
|---|---|
| Median age (IQR) | 49 (16.25) |
| Male[b] | 26 (65%) |
| Hispanic | 13 (33%) |
| African American | 8 (20%) |
| Caucasian | 19 (48%) |
| Uninsured | 34 (85%) |
| Medicaid/Medicare | 2 (5%) |
| Unstable living srrangement | 24 (60%) |
| English as preferred language | 34 (85%) |
| IV drug use | 23 (58%) |
| | |
| Comorbidities | |
| Substance use disorder[c] | 25 (63%) |
| Diabetes | 15 (38%) |
| HCV | 12 (30%) |
| Alcohol use disorder | 6 (15%) |
| Congestive heart failure[d] | 6 (15%) |
| Schizophrenia or bipolar disorder | 5 (13%) |
| HIV | 4 (10%) |
| Cirrhosis[e] | 4 (10%) |
| CKD | 3 (8%) |
| HBV | 1 (3%) |
| Dementia | 1 (3%) |

[a]IQR, interquartile range; IV, intravenous; HIV, human immunodeficiency virus; HCV, hepatitis C virus; HBV, hepatitis B virus; CKD, chronic kidney disease.
[b]Data are presented as counts (%).
[c]Includes nonintravenous drug use, excluding cannabis.
[d]Heart failure with a preserved or reduced ejection fraction.
[e]All from chronic hepatitis C or alcohol.

during the infusion, which improved with a reduction of the infusion rate. Delayed adverse event monitoring was limited due to the low follow-up rates in the clinic. Only 15 (37.5%) of the included patients were seen in follow-up clinic visits.

The concomitant use of other antimicrobials was seen in 6 (15%) patients. The agents used included metronidazole (n = 3, 7.5%), ciprofloxacin (n = 2, 5%), cefuroxime (n = 1,

**TABLE 2** Clinical indications, pathogens, and dosing strategies

| Clinical indications[a] | Value N = 40 |
|---|---|
| ABSSTI | 18 (45%) |
| BSI | 27 (67.5%) |
| Osteomyelitis | 16 (40%) |
| IE | 4 (10%) |
| Septic arthritis | 4 (10%) |
| | |
| Pathogens[b] | |
| MRSA | 23 (57.5%) |
| MSSA | 12 (30%) |
| *Streptococcus* species | 7 (17.5%) |
| CoNS | 5 (12.5%) |
| Enterococcus species | 1 (2.5%) |
| | |
| Dalbavancin dosing strategies | |
| 1,500 mg IV once | 27 (67.5%) |
| 1,125 mg IV once[c] | 1 (2.5%) |
| 1,500 mg IV weekly ×2 doses[d] | 12 (30%) |
| Outpatient doses[e] | 13 (32.5%) |

[a]ABSSTI, acute bacterial skin and soft tissue infection; BSI, bloodstream infection; IE, infective endocarditis; MRSA, methicillin-resistant *Staphylococcus aureus*; MSSA, methicillin-sensitive *Staphylococcus aureus*; CoNS, coagulase negative staphylococcus; IV, intravenous.
[b]11 or 27.5% of patients had polymicrobial infections.
[c]Renally adjusted dosing.
[d]Given on day 1 and day 8.
[e]Four patients were treated in the outpatient setting only.

**TABLE 3** Characteristics of patients with treatment failure[a]

| Age | Clinical indication | Pathogen | Dosing | Comment |
|-----|---------------------|----------|--------|---------|
| 30 | BSI with abscess | MRSA | 1,500 mg ×1[b] | Recurrence of purulent cellulitis with abscess due to intramuscular heroin injections into wound |
| 34 | OM, ABSSTI | MSSA | 1,500 mg weekly ×2[c] | Assault victim that self-discharged 3 times without indicated surgical source control |
| 52 | ABSSTI, BSI, OM | MRSA | 1,500 mg ×1[b] | Multiple self-discharges during which the patient declined the removal of the retained needle and incision and drainage |
| 55 | ABSSTI with BSI | MRSA | 1,500 mg ×1[b] | Burn victim with acute on chronic worsening of wounds, unclear if recurrence or new infection |
| 50 | BSI | CoNS[d] | 1,500 mg ×1[b] | Readmitted with concern for septic arthritis of the ankle |

[a]BSI, bloodstream indications; MRSA, methicillin-resistant *Staphylococcus aureus*; OM, osteomyelitis; ABSSTI, acute bacterial skin and soft tissue infections; MSSA, methicillin-sensitive *Staphylococcus aureus*, AMA, against medical advice; CoNS, coagulase negative *Staphylococcus aureus*.
[b]Received initial treatment course with IV vancomycin prior to dalbavancin.
[c]Received cefazolin prior to dalbavancin.
[d]4 species of CoNS.

2.5%), amoxicillin-clavulanic acid (*n* = 1, 2.5%), and intravenous piperacillin-tazobactam (*n* = 1, 2.5%). One patient received suppressive doxycycline after the dalbavancin therapy, given the paraspinal and epidural abscess that was still present on imaging at the follow-up.

Deviations from the dalbavancin established protocol were seen in this study and were determined to be necessary by the consulting ID physician at the time. This was often necessitated by self-discharges and by the patient declining a transfer to a skilled nursing facility.

**Primary/secondary outcomes.** A total of 5 out of the 40 patients (12.5%) were deemed to have a clinical failure. Clinical failures were attributed to either self-discharging prior to source control, inadequate outpatient wound care, or ongoing intravenous/intramuscular drug use. The specific circumstances for the clinical failures can be seen in Table 3, and the percentage of clinical failures by clinical indication can be seen in Fig. 1. The incidence of all-cause emergency department (ED) visits and readmissions at 0 to 30, 31 to 60, and 61 to 90 days were 16, 10, and 12, respectively. These noninfection-related visits and readmissions were due to alcohol withdrawal, heart failure exacerbation, psychiatric complications, recurrent pulmonary embolism from anticoagulation nonadherence, HIV medication refill, constipation, hyponatremia complications, chronic pain control, and trauma.

At the end of the review period, the electronic medical record was reviewed. A total of 3 patients had died, 2 of whom died over 1 year after the administration of dalbavancin

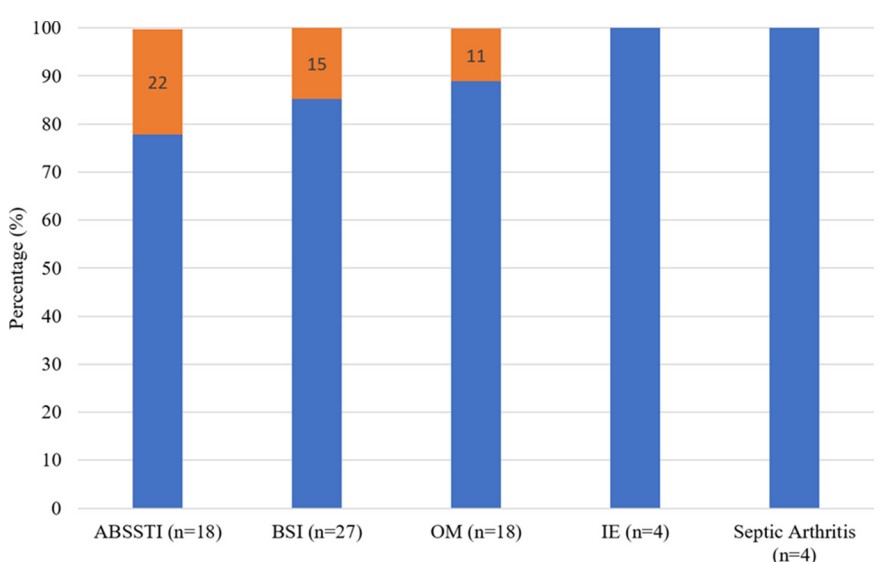

**FIG 1** Clinical failure by indication. 4 out of 5 patients with more than 1 indication.

and the last of whom died within 3 months of administration. The two patients who were over a year out from treatment died from cardiogenic shock and suicide, respectively. The death of the patient who died within 3 months of treatment was attributed to complications from dementia and/or a chronic sacral decubitus ulcer.

Given that all of the patients did not qualify for traditional OPAT, the summation of projected length of treatment was determined to be the total inpatient, rehabilitation, and nursing home days required if the patient were to receive the standard of care (SOC), a total of 998 days. The actual total length of stay for all patients was then subtracted from that amount, resulting in 566 saved days. The mean length of stay was 12 days ($\pm8.13$ [standard deviation]), with an average of 14.51 projected days saved ($\pm11.26$ [standard deviation]). This resulted in a total of $1,208,410 saved, based on Parkland's estimated charge data for a single medical-surgical bed (multiplied by the 566 days saved). This excludes the cost of medications for the SOC. This estimate is based on the cost of a hospital bed and nursing for 1 day only. Thus, it is likely grossly underestimated.

## DISCUSSION

To our knowledge, this is the largest study investigating the use of dalbavancin in complicated Gram-positive infections in a high-risk population. Compared to previously published clinical trials and retrospective studies conducted primarily in Europe, this study includes PWID, those with unstable living conditions, and a more racially diverse population. Our study also shows a higher percentage of study patients with *S. aureus* infections (3, 4, 12). As seen in a study by Bryson-Cahn et al., we observed a likely adequate treatment response in a population with a heavy percentage of PWID, such as ours, with limited follow-up available. However, our study includes a wide range of Gram-positive pathogens, not just *S. aureus*, and it had 100% treatment completion (not including surgical source control) (12). Our findings suggest that the use of long-acting, second-generation lipoglycopeptides for off-label use for complicated Gram-positive infections is a safe and reasonable alternative in this patient population, which would not qualify for traditional OPAT. This results in a reduction of the morbidity/mortality of peripheral inserted central catheter (PICC) placement, prolonged intravenous antibiotics, and prolonged inpatient or facility stays. The importance of this is particularly poignant during the current COVID-19 pandemic, as our health care system is faced with limited resources and hospital beds as well as the need to avoid unnecessary health care exposure in order to prevent transmission. In addition, those over 65 years old are at a greater risk of complications of COVID-19 and often have difficulty affording IV medications under certain Medicare plans, thereby demonstrating a need for alternatives, such as dalbavancin.

Invasive Gram-positive infections are often treated in an outpatient setting through OPAT programs (15). However, in practice, this is not feasible for those who have unstable living conditions or active IDU. Homelessness and IDU often occur concomitantly, making outpatient treatment plans difficult. Thus, these patients often remain in the hospital or are sent to outside facilities to complete therapy. Often, patients self-discharge from skilled nursing facilities or rehabilitation centers without the treating provider's knowledge, have inadequate laboratory drug monitoring, and develop hospital-acquired infections (16). Given the low rates of follow-up in our patient population, we defined clinical failure as ED or inpatient readmission over a 90-day period, given the inability to track other objective variables and the ability to track health care usage throughout the metroplex. We were able to screen ED visits and readmissions at all local hospitals, with one exception. Our data show similar rates of clinical failure (12.5%) to those of previously published studies of invasive Gram-positive infections (3–6, 8, 17). The use of dalbavancin mitigates the risk of oral regimens when IV therapy is indicated, adverse drug events from the SOC, and the early termination of therapy in this population. Dalbavancin also ensures adherence in circumstances in which oral therapy adherence cannot be guaranteed.

The benefit of OPAT programs has clearly been shown in the literature, resulting in proven safety, clinical efficacy, improved patient satisfaction, and cost-effectiveness

(18–19). The COVID-19 pandemic has also required innovative ways by which to discharge stable patients, reduce the census at nursing/rehabilitation facilities, and reduce health care exposure for patients. Often, the underlying comorbidities that resulted in the indication for IV antibiotics predisposed the patient to severe COVID-19 (20). Given this risk of severe COVID-19, the infectious diseases division at our institution was motivated to increase the usage of off-label dalbavancin. In our study, in addition to the cost-savings, dalbavancin resulted in a reduction of 566 or 57% of the total projected inpatient or facility days, which, in midst of the COVID-19 pandemic, helped to free up essential beds and to reduce health care exposure.

The use of long-acting, second-generation lipoglycopeptides offers a unique tool by which to reduce health care exposure for invasive Gram-positive infections that have historically required prolonged IV antibiotics, PICC placement, and often prolonged admissions or facility stays for high-risk patients. The shift of care to the outpatient setting has been highlighted by the COVID-19 pandemic and rising health care costs. A barrier to usage has always been cost, as was recently described in one recent retrospective study by Gonzales et al., which showed a total cost per patient of $4,770 (21). At safety net hospitals, such as Parkland, this cost is circumvented with 340B pricing and drug-company assistance programs. A compelling argument can be made that the cost of the drug is negated by the savings derived from the reduction in inpatient days, facility days, central venous catheter complications, and the need for home health services.

There are several limitations to our study. As is common in this high-risk population, there are limitations on adequate follow-ups, resulting in a lack of objective data with which to confirm clinical cures. 62.5% of the patients either did not show up for their option appointments or did not schedule appointments prior to discharge. Currently, there is a lack of randomized controlled trials investigating the use of dalbavancin in cases of BSI, IE, prosthetic joint infection, vertebral OM, and central nervous system infections. However, there are ongoing clinical trials to fill these gaps, such as the Dalbavancin as an Option for Treatment of *Staphylococcus Aureus* Bacteremia or DOTS trial. It was also difficult to assess the effectiveness of the initial antibiotic course, compared to the added benefit of dalbavancin. Resistance testing was also unavailable. So, in the cases of clinical failure, we were unable to determine whether this was a result of induced resistance. Less than 50% of our patients had surgical source control, which, in theory, could result in inducible resistance, given the persistent nidus of infection and the drug concentrations dropping below the minimum inhibitory concentration (MIC) over time (22–23).

In summary, our study shows that dalbavancin is a reasonable treatment option for invasive Gram-positive infections that require prolonged IV antibiotics in those who do not qualify for traditional OPAT or those who seek to reduce health care exposure during the COVID-19 pandemic. Additional prospective studies, including studies on this high-risk population, are needed to validate this treatment strategy.

## MATERIALS AND METHODS

This was a retrospective observational study performed from July of 2019 to August of 2021 at Parkland Health, an 836-bed safety net hospital that is located in Dallas, Texas. It is affiliated with the University of Texas Southwestern (UTSW) Medical System. The study received approval from the UTSW Institutional Review Board with a waiver of consent.

A protocol by which to assess patients' eligibility for dalbavancin was created to streamline decisions between dalbavancin versus an alternative method to complete OPAT. Per the protocol, infections eligible for dalbavancin included ABSSTIs, OM, septic arthritis, BSI, and native-valve IE due to suspected or confirmed Gram-positive infections. Conditions outside the scope of the protocol included central nervous system infections, including epidural abscesses, uncontrolled abscesses/collections, patients with prosthetic material, including but not limited to prosthetic heart valves, orthopedic hardware, pacemakers, pumps, and brain stimulators, and patients receiving hemodialysis. Patients were excluded from consideration for dalbavancin if they had a history of allergic reaction to dalbavancin or oritavancin, a history of IgE-mediated reaction to vancomycin, were pregnant or nursing, or had Child-Pugh Class B or C liver disease. The OPAT pharmacist or ID consult team made recommendations for dalbavancin, as guided by the protocol, and assessed patients on a case by case basis. All of the patients who received dalbavancin from July of 2019 to August of 2021 were included in the study, regardless of indication. The primary outcome assessed was clinical failure, a composite outcome that was defined by ED visits or

hospital readmissions at 30, 60, or 90 days, relating to the original infection. Given the difficulty with follow-ups in this population, this was used as a surrogate marker. Secondary outcomes included adverse drug events (ADE), all-cause ED visits or hospital readmissions, and the total number of inpatient, rehabilitation, and nursing facility days saved. The number of days saved was calculated using the minimal duration of treatment with the standard of care that the patient would have required, as they were not traditional OPAT candidates.

Data were collected from the electronic medical record and were stored in a secure internal server. Records were retrospectively reviewed, and the following data were collected: age, race, past medical history, housing status, IDU history, insurance status, inpatient/outpatient setting, length of stay, projected duration of therapy, time to follow-up, and return to ED or readmission within 30, 60, and 90 days. To capture ED visits or readmissions outside our health system, the Care-Everywhere function in Epic was used to screen encounters in all of the major hospital systems in the Dallas metroplex, with the exception of one hospital system. Parkland is a safety net hospital. Thus, most patients return to Parkland if there are ongoing medical needs. Regarding infection and treatment characteristics, the following were collected: the type of infection, pathogen isolated, treatment regimen (other than dalbavancin), surgical management, adverse drug events, number of dalbavancin doses, and location of the dalbavancin infusion. The projected length of treatment was determined by an infectious diseases specialist at the time of consultation.

Descriptive analyses were performed for all of the data points using the Microsoft Excel software package. The absolute and relative frequencies (%) were calculated for qualitative variables, and means and standard deviations were calculated for quantitative variables. A separate analysis was conducted to estimate the hospital, nursing home, or rehabilitation days saved as well as the resulting cost-savings, using Parkland's estimated charge data for a single medical-surgical bed and the cost to Parkland for outpatient skilled nursing facilities/rehabilitation centers.

## ACKNOWLEDGMENTS

We acknowledge the infectious diseases faculty and fellows at the University of Texas Southwestern Health System as well as the referring physicians, patients, and department of pharmacy at the Parkland Health & Hospital System.

This research did not receive any funding from the institution or any specific grant funding from the public, commercial, or nonprofit sectors.

All authors contributed significantly to the work presented and to the preparation of the manuscript. All authors have read and approved the final submission.

No author has any conflicts of interest to report. The data was presented as part of the IDweek 2021 Virtual Conference in September to October of 2021.

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
