## [Reviewer comments · Microbiology Spectrum]

Microbiology Spectrum

Evaluation of Dalbavancin Use on Clinical Outcomes, Cost-Savings, and Adherence at a Large Safety Net Hospital

Richard Lueking, Wenjing Wei, Norman Mang, Jessica Ortwine, and Jessica Meisner

Corresponding Author(s): Richard Lueking, University of Texas Southwestern Medical Center, and Jessica Meisner, University of Pennsylvania Perelman School of Medicine

Review Timeline:

Submission Date:	June 23, 2022
Editorial Decision:	August 16, 2022
Revision Received:	October 19, 2022
Accepted:	November 18, 2022

Editor: Adriana Rosato

Reviewer(s): Disclosure of reviewer identity is with reference to reviewer comments included in decision letter(s). The following individuals involved in review of your submission have agreed to reveal their identity: Matthew P Crotty (Reviewer #1)

Transaction Report:

DOI: <https://doi.org/10.1128/spectrum.02385-22>

August 16, 2022

Dr. Richard Lueking
University of Texas Southwestern Medical Center
Infectious Diseases
5323 Harry Hines Blvd
Dallas, Texas

Re: Spectrum02385-22 (Evaluation of the Impact of Dalbavancin Usage on Clinical Outcomes, Cost-Savings, and Adherence at a Large Safety Net Hospital)

Dear Dr. Richard Lueking:

Thank you for submitting your manuscript to Microbiology Spectrum. The manuscript has been reviewed by two experts in the field. I concur with their assessment, therefore I will strongly suggest to the authors to carefully address their comments.

Link Not Available

Sincerely,

Adriana Rosato

Journals Department
Reviewer comments:

Reviewer #1 (Comments for the Author):

General comments

1. Much of the results section is quite repetitive with the tables and figures. Consider being more succinct and focused in verbiage utilized to describe results and remove content which is duplicated in tables/figures.
2. With LOS on average being 12 +/- 8 within 1 SD: many patients appear to have been nearly (or completely treated) for an adequate duration as inpatients (e.g., ABSSI with bacteremia may require 14-17 days of therapy which falls within the range of many patients LOS). A description of antibiotic therapy received prior to dalbavancin appears necessary to adequately understand the efficacy described in this patient cohort.

3. Because many patients had prolonged hospitalizations prior to dalbavancin, a detailed investigation of antibiotic therapy prior is needed to contextualize the results. How were patients deemed to be 'stable/safe/ready for dalbavancin'? It is unclear how much dalbavancin administration truly impacted outcomes of patients if treated for multiple weeks with other antibiotics prior to that point.
4. Without susceptibility testing conducted on any isolates some address for resistance and the possible impact of prolonged exposure (at eventually sub-inhibitory concentrations) should be a consideration for discussion.

Specific comments

1. Title: the title does not seem to reflect the content of the manuscript. Without a comparator group, 'impact' is difficult to glean. Additionally, adherence is also not a focal point of the manuscript nor is it something that can be adequately evaluated without a comparator for this particular agent (i.e., one-time or 2-dose regimen with dalbavancin are unlikely to have "adherence" issues which could be beneficial in the patient population described but no real insight is able to be determined without comparison to a 'standard care' group). Consider revising as a description in a real-world setting.
2. Intro Lines 83-90: consider streamlining this section to be more forthright with the purpose and need of the study. Currently reads overly elaborate for its purpose in this section.
3. Line 115-116: Was Care Everywhere function specifically utilized for the purpose of determining whether patients were readmitted to [a] hospital within 30, 60, 90 days? Please be more deliberate in stating.
4. Line 123: "previously published healthcare economic data" is a vague statement left unreferenced. Please revise to state what specifically the data in reference is and add whatever citation it refers to.
5. Lines 98 and 151, table 1: Although retained hardware, epidural abscesses, and cirrhosis are mentioned as exclusions from use of dalbavancin in institutional protocol, there appear to have been protocol deviations described. Please consider describing what the process for approval is and how deviations from the protocol were determined to be acceptable. This is likely to be challenges that other hospitals will face, so the description may prove helpful.
6. Line 156: "rates" is probably not the correct word to use here as the raw numbers are presented. Consider changing that word or describing numbers as percentage of patients. It appears 40% of patients were readmitted within 30 days (16 of 40 patients). Please provide the number of instances for each cause of readmission (lines 158-160) to better understand these high rates of readmission.
7. Lines 198-199: why is oral therapy considered inferior? There are several studies suggesting linezolid and other oral combinations are viable treatment options for bloodstream infections including infective endocarditis and osteoarticular infections. Please consider revising this and expanding discussion of oral therapy as an alternative treatment option to lipoglycopeptides.
8. Lines 206-207: this statement and conclusion seems like a large jump from the data described here. Without evaluation of local transmission at the time of dalbavancin administration much less the rates of transmission in the hospital (and a host of other factors) this statement is not appropriate from the data described. Please revise. Consider something along the lines of describing at times when beds were of critical/dire need.
9. Line 225: this appears to be new results being presented in the discussion. Please more fully describe source control or lack thereof more thoroughly in results section.
10. Figure 2 is unnecessary and duplicative.

Reviewer #2 (Comments for the Author):

Introduction

85 Disagree with authors that previously published work on real-life use of dalbavancin does not include PWID or houseless patients as these populations are largely where the benefit of long-acting glycopeptides are seen, as described in a number of similar retrospective reviews. Recommend rephrasing.

Methods

Could rate of COVID-19 infection be reported as that is noted by the authors as an increased motivation for dalbavancin use during the study period?

Methods should include description of how cost calculation was conducted. Specifically, how were the cost of saved days calculated, were separate costs calculated for inpatient days vs nursing home days or rehabilitation days?

Results

165 - Definition of projected length of therapy should be included in methods rather than results.

Discussion

This could be expanded to describe further complications in COVID-19 if that's the argument the authors wish to make. For example, patients over the age of 65 are both at higher risk of severe disease and most often insured by Medicare which does not cover home infusion for antibiotics, thus creating even more need for alternatives.

Would caution against drawing conclusions about clinical failure given small sample, high loss to follow up and variety of infectious indications treated.

Figure 2 - the pie chart is unnecessary for understanding of the days saved results

Staff Comments:

Preparing Revision Guidelines

Please return the manuscript within 60 days; if you cannot complete the modification within this time period, please contact me. If you do not wish to modify the manuscript and prefer to submit it to another journal, please notify me of your decision immediately so that the manuscript may be formally withdrawn from consideration by Microbiology Spectrum.

Dear Editor:

On behalf of the authors we would like to thank the editors and reviewers for their time and effort in critically assessing this work and facilitating an expedited review of our manuscript "Evaluation of Dalbavancin Use on Clinical Outcomes, Cost-Savings, and Adherence at a Large Safety Net Hospital."

We have carefully considered each comment and are submitting a revised version of the manuscript that encompasses changes as suggested by the reviewers. The marked-up version of the revision provided as a supplemental file named "Dalbavancin Manuscript_marked up version."

Reviewer #1 (Comments for the Author):

General comments

1. Much of the results section is quite repetitive with the tables and figures. Consider being more succinct and focused in verbiage utilized to describe results and remove content which is duplicated in tables/figures.

As recommended, we have edited the results section to be less repetitive and have eliminated figure #2.

2. With LOS on average being 12 +/- 8 within 1 SD: many patients appear to have been nearly (or completely treated) for an adequate duration as inpatients (e.g., ABSSI with bacteremia may require 14-17 days of therapy which falls within the range of many patients LOS). A description of antibiotic therapy received prior to dalbavancin appears necessary to adequately understand the efficacy described in this patient cohort.

Patient's received standard pf care for varying amounts of time prior to dalbavancin administration. 87.5% of patients had staph aureus and very few meet criteria for uncomplicated bacteremia course of 14 days as determined by the ID consultant at the time. As described in the results, an average of 14 days was saved by administration of dalbavancin based on the length of treatment needed determined by the ID consultant.

3. Because many patients had prolonged hospitalizations prior to dalbavancin, a detailed investigation of antibiotic therapy prior is needed to contextualize the results. How were patients deemed to be 'stable/safe/ready for dalbavancin? It is unclear how much dalbavancin administration truly impacted outcomes of patients if treated for multiple weeks with other antibiotics prior to that point.

All patient received standard of care antimicrobials prior to switching to dalbavancin. The infectious diseases specialist was the ultimate determinate of when patients were able to be switched to dalbavancin. In cases of bacteremia, this was upon culture clearance, with the exception of patients who self-discharged.

4. Without susceptibility testing conducted on any isolates some address for resistance and the possible impact of prolonged exposure (at eventually sub-inhibitory concentrations) should be a consideration for discussion.

This is addressed in the second-to-last paragraph of the discussion section regarding limitations to our study.

Specific comments

1. Title: the title does not seem to reflect the content of the manuscript. Without a comparator group, 'impact' is difficult to glean. Additionally, adherence is also not a focal point of the manuscript nor is it something that can be adequately evaluated without a comparator for this particular agent (i.e., one-time or 2-dose regimen with dalbavancin are unlikely to have "adherence" issues which could be beneficial in the patient population described but no real insight is able to be determined without comparison to a 'standard care' group). Consider revising as a description in a real-world setting.

Appreciate the reviewer's comment, have revised the title to read as "Evaluation of Dalbavancin Use on Clinical Outcomes, Cost-Savings, and Adherence at a Large Safety Net Hospital."

2. Intro Lines 83-90: consider streamlining this section to be more forthright with the purpose and need of the study. Currently reads overly elaborate for its purpose in this section.

As recommended, have eliminated a portion of this paragraph to be more concise.

3. Line 115-116: Was Care Everywhere function specifically utilized for the purpose of determining whether patients were readmitted to [a] hospital within 30, 60, 90 days? Please be more deliberate in stating.

This statement was edited to clarify that the care-everywhere function was used to capture data points (ED visits or readmission at 30,60,90 days) from those whom sought care at another facility after dalbavancin administration at Parkland.

4. Line 123: "previously published healthcare economic data" is a vague statement left unreferenced. Please revise to state what specifically the data in reference is and add whatever citation it refers to.

As recommended, this was revised to state that cost savings were calculated by on the daily charge data for a single medical-surgical bed at Parkland Hospital.

5. Lines 98 and 151, table 1: Although retained hardware, epidural abscesses, and cirrhosis are mentioned as exclusions from use of dalbavancin in institutional protocol, there appear to have been protocol deviations described. Please consider describing what the process for approval is and how deviations from the protocol were determined to be acceptable. This is likely to be challenges that other hospitals will face, so the description may prove helpful.

We thank the reviewer for pointing out this discrepancy. We have added a clarifying statement that states that the consulting ID physician could breach the protocol if determined to be clinical indicated.

6. Line 156: "rates" is probably not the correct word to use here as the raw numbers are presented. Consider changing that word or describing numbers as percentage of patients. It appears 40% of patients were readmitted within 30 days (16 of 40 patients). Please provide the number of instances for each cause of readmission (lines 158-160) to better understand these high rates of readmission.

As recommended, have changed rates to incidence to better describe the data. This data reflects non-related ED visits and readmissions and thus incidence of each indication was not provided.

7. Lines 198-199: why is oral therapy considered inferior? There are several studies suggesting linezolid and other oral combinations are viable treatment options for bloodstream infections including infective endocarditis and osteoarticular infections. Please consider revising this and expanding discussion of oral therapy as an alternative treatment option to lipoglycopeptides.

This statement has been clarified as to not indicate IV therapy is superior to oral therapy in all cases. Rather in situations where IV therapy is required or adherence to PO regimen is in question, dalbavancin is a reasonable alternative.

8. Lines 206-207: this statement and conclusion seems like a large jump from the data described here. Without evaluation of local transmission at the time of dalbavancin administration much less the rates of transmission in the hospital (and a host of other factors) this statement is not appropriate from the data described. Please revise. Consider something along the lines of describing at times when beds were of critical/dire need.

As recommended, we have removed this statement and revised it to say that by reducing length of stays we were able to free up essential beds needed during the COVID-19 pandemic.

9. Line 225: this appears to be new results being presented in the discussion. Please more fully describe source control or lack thereof more thoroughly in results section.

A statement has been added to the results section as reviewer pointed out that it had not been discussed previously.

10. Figure 2 is unnecessary and duplicative.

As recommended, figure 2 has been removed

Reviewer #2 (Comments for the Author):

Introduction

85 Disagree with authors that previously published work on real-life use of dalbavancin does not include PWID or houseless patients as these populations are largely where the benefit of long-acting glycopeptides are seen, as described in a number of similar retrospective reviews. Recommend rephrasing.

We agree with the reviewers comment and have clarified the statement in the introduction to state the PWID were not likely to be included in the early European studies of dalbavancin.

Methods

Could rate of COVID-19 infection be reported as that is noted by the authors as an increased motivation for dalbavancin use during the study period?

Unfortunately, we do not have access to this data. Agree with the reviewer that this information would have been valuable. Our hospital is the only safety net hospital in North Texas and saw 40% of all COVID cases for Dallas. This was a vital point in our decision to expand our dalbavancin protocol.

Methods should include description of how cost calculation was conducted. Specifically, how were the cost of saved days calculated, were separate costs calculated for inpatient days vs nursing home days or rehabilitation days?

As recommended, a statement was added to clarify that the cost savings was calculated based on data provided by the Parkland business office. This included the average charge-cost for a single medical-surgical bed. Since parkland is a safety-net hospital, the health system pays for rehabilitation/SNF stays for uninsured patients.

Results

165 - Definition of projected length of therapy should be included in methods rather than results.

This was moved to the methods section as recommended.

Discussion

This could be expanded to describe further complications in COVID-19 if that's the argument the authors wish to make. For example, patients over the age of 65 are both at higher risk of severe disease and most often insured by Medicare which does not cover home infusion for antibiotics, thus creating even more need for alternatives.

Agree with reviewer that his comment strengths the argument we are trying to make. Addition comment has been added in the discussion section.

Would caution against drawing conclusions about clinical failure given small sample, high loss to follow up and variety of infectious indications treated.

Our conclusion is that Dalbavancin seems to be a reasonable and safe alternative in this patient population. We refrain from stating superiority/non-inferiority as the study was not designed to show that.

Figure 2 - the pie chart is unnecessary for understanding of the days saved results

As recommended, figure 2 has been removed

November 10, 2022

Dr. Richard Lueking
University of Texas Southwestern Medical Center
Infectious Diseases
5323 Harry Hines Blvd
Dallas, Texas

Re: Spectrum02385-22R1 (Evaluation of Dalbavancin Use on Clinical Outcomes, Cost-Savings, and Adherence at a Large Safety Net Hospital)

Dear Dr. Richard Lueking:

Your manuscript has been accepted, and I am forwarding it to the ASM Journals Department for publication. You will be notified when your proofs are ready to be viewed.

Sincerely,

Adriana Rosato
Editor, Microbiology Spectrum
